# Integrative Multi-Omics Analyses Reveal the Global Regulation Network of the Microalga *Nannochloropsis oceanica* Under Nitrogen Stress Adaptation

**DOI:** 10.3390/biology14111599

**Published:** 2025-11-15

**Authors:** Wuxin You, Can Xu, Jingyi Zhang, Ansgar Poetsch

**Affiliations:** 1Core Facility for Biomedical Science, Nanchang University, Nanchang 330031, China; youwuxin@ncu.edu.cn; 2School of Basic Medical Science, Institute of Biomedical Innovation, Jiangxi Medical College, Nanchang University, Nanchang 330031, China; canxu@ncu.edu.cn; 3Queen Mary School, Jiangxi Medical College, Nanchang University, Nanchang 330031, China; qingfox01@163.com; 4Plant Biochemistry, Ruhr University Bochum, 44801 Bochum, Germany

**Keywords:** triacylglycerols, post-translational modification, multi-omics, lipid biosynthesis

## Abstract

Microalgae called *Nannochloropsis oceanica* turn sunlight and carbon dioxide into valuable oil, which has huge potential for bio-energy application, but only when exposed to nitrogen-limiting conditions. How they sense the nitrogen shortage and trigger oil accumulation remains unclear. For this, the cellular protein network was examined with a focus on protein phosphorylation, since it is well known that decoration of certain sites in proteins with phosphate affects their individual function and potentially the whole network. In total, 1371 phosphorylation sites on more than 800 proteins were identified. Two clear waves appeared in the network: first the cells saved nitrogen, then they boosted the oil-producing machinery. Moreover, phosphorylation of the Target of Rapamycin (TOR) signaling pathway, which is a master growth control pathway, was regulated, suggesting that this pathway was the nitrogen sensor. These results support future design of strains to become better oil producers to supply more and cheaper renewable fuel towards a greener economy.

## 1. Introduction

*Nannochloropsis oceanica* is an eukaryotic microalga of the *Eustigmatophyceae* class and the *Nannochloropsis* genus, sharing an evolutionary lineage with diatoms and brown algae. *Nannochloropsis* occupies a pivotal role in aquatic ecosystems [1]. Ubiquitously distributed in marine and freshwater environments, these microalgae significantly contribute to the carbon cycle as primary producers, harnessing atmospheric CO_2_ through photosynthesis. With their exceptional capacity to rapidly synthesize triglycerides, serving as repositories of solar energy and sequestered carbon, *Nannochloropsis* shows bright prospects for bio-energy applications [2,3]. Furthermore, their potential to mitigate industrial greenhouse gas emissions through CO_2_ assimilation bolsters their ecological and economic relevance. *Nannochloropsis* also synthesizes nutritionally valuable polyunsaturated fatty acids, such as eicosatetraenoic acid. Their compact haploid genomes, which are amenable to genetic manipulation, have established them as prime models for the synthetic biology of eukaryotic microalga species, facilitating strain engineering and biotechnological advancements [4].

Despite these traits, *Nannochloropsis* lacks a competitive edge in the production of industrial biofuels [5,6]. To achieve this, a profound investigation of its biology from the molecular to the system level is crucial [7,8]. Current strain engineering focuses primarily on the modulation of gene expression through over-expression or silencing strategies, often overlooking complex regulatory mechanisms that operate beyond transcriptional control. Post-translational modifications (PTMs) play crucial roles in determining protein function, stability, localization, and interaction dynamics in adaptive responses, yet these aspects remain largely underexplored in microalgae [9,10,11]. PTMs, which enzymatically or spontaneously alter protein structures and functional features [12], are involved in the regulation of cellular metabolism and enzyme activities. Phosphorylation is a ubiquitous form of PTM in cells [12,13]. Extensive research has revealed the central role of protein phosphorylation in glycolysis, the TCA cycle, carbon fixation, and lipid metabolism pathways [14,15]. For instance, the regulation of pyruvate dehydrogenase activity and TAG synthesis in *Nannochloropsis salina* has been directly attributed to phosphorylation events catalyzed by a specific kinase [16]. To achieve this, a profound investigation of its biology from the molecular to the system level is crucial.

However, compared to other microalgae such as *Chlamydomonas* and *Phaeodactylum*, where PTM roles are better characterized, the specific mechanisms of phosphorylation in *Nannochloropsis*, particularly under nitrogen stress, are less understood. Compared to *Chlamydomonas* and *Phaeodactylum*, research on post-translational modifications (PTMs) in *Nannochloropsis* is significantly less advanced, particularly in terms of comprehensive proteomic results and functional characterization of PTM-related enzymes [17,18]. While *Chlamydomonas* and *Phaeodactylum* benefit from extensive proteomic mapping and established genetic tools to delineate PTM roles in photosynthesis, metabolism, and stress responses, PTM studies around *Nannochloropsis* are primarily in nascent stages, focusing on isolated modifications like lactylation under specific conditions [17]. This creates a substantial gap in understanding how PTMs regulate *Nannochloropsis* growth, stress adaptation, and lipid accumulation, which are crucial for its biotechnological potential [19,20,21]. Bridging this gap requires broader proteomic profiling and in-depth functional studies of PTMs by using advanced genetic and analytical tools, akin to those employed in other well-studied microalgae [22].

Despite the established role of phosphorylation in the regulation of pyruvate dehydrogenase activity in *Nannochloropsis*, a broader understanding of how protein phosphorylation signals orchestrates nitrogen starvation, particularly in the context of TAG accumulation in this genus remains elusive [23,24,25,26].

The gap is critical because nitrogen starvation commonly triggers lipid production in *Nannochloropsis*, yet the underlying phosphorylation-mediated regulatory networks are poorly defined. Recent research on the regulation of protein phosphorylation has focused on the Target of Rapamycin (TOR) kinase pathway, given its conserved function as a master regulator of growth and metabolism in diverse organisms, including microalgae. In particular, studies have shown that TOR kinase inhibitors, when applied to species such as *Phaeodactylum tricornutum*, result in the accumulation of TAG [27,28], implicating TOR signaling as a potential mechanism for manipulating lipid biosynthesis in algae. Furthermore, previous analyses of TOR phosphorylation patterns under different nutrition regimes exemplify a refined and specific regulation of TOR in higher plants; certain sites were found to be sensitive to nitrogen, while others responded to sugars or amino acids, implying that TOR integrates different signals through phosphorylation modes [29,30].

Based on this analogy and preliminary evidence from related microalgae, we hypothesize that TOR signaling may be a key regulator in the nitrogen stress response of *Nannochloropsis*, though the exact mechanism remains a research question. Recent studies have shown that when TOR kinase inhibitors are applied to species such as *Phaeodactylum tricornutum*, they lead to the accumulation of TAG, suggesting that TOR signaling may be a potential mechanism regulating lipid biosynthesis in algae [29]. Furthermore, previous analyses of TOR phosphorylation patterns under different nutritional schemes in advanced plants have also exemplified the fine and specific regulation of TOR; it was found that certain sites are sensitive to nitrogen, while others respond to sugar or amino acids, indicating TOR integrates different signals through variant phosphorylation patterns [29]. The TOR kinase pathway is highly conserved in eukaryotes and widely recognized as a master regulator of cell growth, metabolism, and stress response [30,31]. Thus, we have reason to believe that the TOR pathway likely plays a central role in sensing the nitrogen starvation and inducing TAG accumulation. Hence, this article attempts to verify whether the TOR signaling pathway in *Nannochloropsis* serves as a key regulatory factor in response to nitrogen starvation by combining its phosphoproteome with proteomic and metabolomic data.

Our previous research on the transcriptomics and proteomics of *Nannochloropsis oceanica* has paved the way for deciphering the temporal adaptation mechanisms to nitrogen depletion and elucidating key metabolic transitions and genes [32]. Several genes with discrepancies between transcripts and protein abundances were found, which can be explained by post-transcriptional regulation mechanisms. This observation motivated us to further investigate the role of protein phosphorylation in adaptation to starvation using phosphoproteomics. In the current study, significant changes have been identified in 126 phosphorylation modifications in metabolic pathways and 9 phosphorylation modifications in the TOR signaling pathway. The results indicated that phosphorylation regulates the metabolism of *Nannochloropsis* under nitrogen-deficient conditions (N-) both directly and indirectly. Furthermore, we found that for many proteins, the phosphor-site occupancies changed earlier than the protein abundance, suggesting an important role for phosphorylation in the detection and initial adaptation to nitrogen deficiency.

## 2. Materials and Methods

### 2.1. Cultivation and Nitrogen Deprivation of *N. oceanica* IMET1

The cultivation of *N. oceanica* IMET1 was optimized and modified previously according to published work [32]; the cultivation of *N. oceanica* IMET1 was performed in three steps. Firstly, cells were suspended in 100 μL f/2 medium (35 g/L sea salt (Real Ocean, New York city, NY, USA), 1 g/L NaNO_3_, 67 mg/L NaH_2_PO_4_·H_2_O, 3.65 mg/L FeCl_3_·6H_2_O, 4.37 mg/L Na_2_EDTA·2H_2_O, trace metal mix (0.0196 mg/L CuSO_4_·5H_2_O, 0.0126 mg/L NaMoO_4_·2H_2_O, 0.044 mg/L ZnSO_4_·7H_2_O, 0.01 mg/L CoCl_2_, and 0.36 mg/L MnCl_2_·4H_2_O, and vitamin mix (2.5 μg/L VB_12_, 2.5 μg/L biotin, and 0.5 μg/L thiamine HCl) from a glycerol storage culture (under sterile conditions) and spread on an agar plate (modified f/2 medium with 1.5% agar). This plate was cultured at 8 μmol photon m^−2^s^−1^, 25 °C without CO_2_ fumigation in a programmable growth chamber (Model SWGC-450, Witeg, Wertheim am Main, Germany) until individual colonies formed. A single colony was removed from this plate and suspended in about 50 mL of f/2 medium and in a shaking flask for a further 5 days (at 50 μmol photons m^−2^s^−1^ 25 °C without CO_2_ fumigation) to an OD_750_ of 5. The culture was pelleted and resuspended in fresh f/2 medium in a 1 L glass tube. The cultivation in the algae tube was carried out under the same conditions as in the growth chamber but with aeration by sterile air. The growth of the culture was monitored spectrophotometrically at OD_750_.

After starting with an OD_750_ of 1.0, six cultures were prepared with modified f/2 medium for 7 days. Subsequently, the cells were pelleted and washed twice in f/2 medium without the nitrogen source (NaNO_3_). The cell pellets were transferred to the new sterile glass tube, resuspended and cultured for a further 10 days under N−. Three biological replicates of algal cultures, corresponding to altogether six column reactors, were cultivated under the N− and N+ conditions, respectively. Cell aliquots were taken at 3, 6, 12, 24, 48, and 10 days from each column by pipette for OD_750_ measurement and proteomic profiling. Bacterial contamination was tested after DAPI staining under a fluorescence microscope at 0 h and 10 days [32]. The samples of each time point were divided into two parts, which were used for proteomics test and the current phosphorylation enrichment test, respectively.

### 2.2. Protein Extraction and Preparation

Cultures were centrifuged to harvest cells, and cell pellets were ground with liquid nitrogen. Lysed cells were suspended in SDS lysis buffer (4% SDS in 0.1 M Tris/HCl pH 7–8) with phosphatase and protease inhibitor cocktail (PhosSTOP®, Roche, Basel, Switzerland). The PhosSTOP® tablet was dissolved in 1 mL ddH2O and mixed with 9 mL lysis buffer. After purification using a C18 stage tip [33], 400 µg proteins were extracted and disulphide bonds were reduced by adding 10 μL of 1 M DTT to the final concentration of 0.05 M at RT for 30 min. After that 10 μL iodoacetamide (550 mM) was added to alkylate cysteines. The proteins were digested with Lys-C at a ratio of 100:1 (*w*/*w*) (Lysyl Endopeptidase®, Wako, Japan). After 3 h, trypsin (Promega®, Madison, WI, USA) was added at a ratio of 100:1 (*w*/*w*).

### 2.3. Phosphorylation-Modified Peptide Enrichment

To identify and quantify the phosphorylated peptides of *Nannochloropsis*, nano class TiO_2_ beads (Titansphere®, GL Sciences, Tokyo, Japan) were employed to enrich the phosphorylated peptides from Lys-C/trypsin-digested *Nannochloropsis* proteins in a ratio of 20:1 (*w*/*w*) to proteins. TiO_2_ beads were suspended in a loading solution (80% ACN, 6% TFA) at a concentration of 100 μL per sample. The TiO_2_ beads were incubated with digested proteins to enrich the phosphorylated peptides. After centrifugation, the beads were washed twice with solvent (60% ACN, 1% TFA) to remove non-specifically bound peptides. The beads were transferred to a C18 Stage Tip and eluted with 30 μL of elution buffer (40% ACN, 15% NH_4_OH,(28%, HPLC grade)). The eluted phosphorylated peptides were transferred to a glass vial and dried with SpeedVac.

### 2.4. LC–ESI-MS/MS Analysis

For the MS analysis of protein digests and the enriched phosphorylated peptides, the same method was used: dried peptides were resuspended in solvent A (0.1% formic acid (FA) in HPLC-grade water (Fisher Scientific GmbH, Schwerte, Germany)) and sonicated for 10 min in an ultrasonic bath (RK-100 H, Heidolph, Schwabach, Germany). The LC-ESI-MS/MS system consisted of a nanoACQUITY gradient UPLC pump (Waters Corporation, Milford, MA, USA) interfaced with an LTQ Orbitrap Elite mass spectrometer (Thermo Fisher Scientific, New York, NY, USA). For the LC, an ACQUITY UPLC 2D VM M-Class Symmetry C18 trap column (100 Å, 5 μm, 180 μm × 20 mm) (Waters Corporation, New York, NY, USA) was coupled to an HSS T3 ACQUITY UPLC M-Class separation column (75 μm × 150 mm) (Waters Corporation, New York, NY, USA). The nanospray source was a PicoTip Emitter Silica Tip (10 μm tip ± 1 μm) (New Objective, New York, NY, USA). Xcalibur (Version 2.2 SP1) was used for the software-based instrument control of the mass spectrometer. For the UPLC method, the flow rate was 0.4 μL/min. A 105 min gradient was used with 0–5 min: 2% solvent B (0.1% formic acid in acetonitrile, UPLC/MS, Fisher Scientific, GmbH, Schwerte, Germany); 5–10 min: 2–5% B; 10–71 min: 5–30% B; 72–77 min: 85% B; 77–105 min: 2% B. The analytical column oven was set to 55 °C, and the heated desolvation capillary was set to 275 °C. A full Orbitrap scan was first performed in the range of 150–2000 *m*/*z* at a resolution of 240,000 MS/MS. The 20 most intensive precursor ions from the full scan were then fragmented using CID (activation time 10 ms and 35% collision energy). The resulting fragments were detected in the ion trap. All precursors of unknown charges or charges ≠ 2 or 3 were rejected for MS/MS analysis.

### 2.5. Identification and Label-Free Quantification of Phosphorylation-Modified Peptides

MaxQuant (Version 1.5.5.1) with the Andromeda search engine was used for phosphorylated peptide identification and label-free quantification (LFQ). The peptides were identified against the complete proteome database of *N. oceanica* IMET1. The minimum score of the modified peptides was 40. The mass tolerance of calibrated precursor ions was set to 4.5 ppm; the mass tolerance of fragment ions was set to 0.6 Da. Only tryptic peptides with up to two missed cleavages were accepted. Phosphorylation of serine, threonine, and tyrosine, oxidation of methionine, and acetylation at protein N-terminus were considered as variable, and carbamidomethylation on cysteine as static peptide modifications. The false discovery rate (FDR) was set to 0.01.

### 2.6. Data Filtering in Perseus

After employing label-free relative quantification (LFQ), 4114 protein sequences were identified and quantified. For the comparison between N− and N+ data, the LFQ-normalized intensities of the samples were log2-transformed using Perseus software (version 1.6.2.2) [34]. For sample comparisons, proteins that were not quantified for at least half of the time points for N− and N+ samples were removed. As for Phosphopeptide analysis, through import mass spectrometry identification results (Phospho (STY) Sites.txt file output from Maxquant) into Perseus. Then filtered the data to removed reverse sequences (decoy database matches), contaminant proteins, and proteins identified only by modification sites. Additionally, filtered phosphorylation sites based on localization probability (>0.75) to ensured accurate site assignment. Furthermore, performed log2 transformation on phosphorylation site intensity values to met the normal distribution requirements for subsequent statistical analysis. Using Z-score method to normalized data across different samples. Subsequently, missing values were replaced using imputation from the normal distribution in Perseus (width 0.3, down shift 1.8). The fold changes between the N− and N+ samples were compared (i.e., log2 LFQ[N−/N+]) using the two-sample *t*-test.

## 3. Results

To characterize the changes in the phosphoproteome during nitrogen starvation in *N. oceanica* IMET1, the abundance of phosphorylated peptides was compared among culture samples taken at 3, 6, 12, 24, 48 h, and 10 days of nitrogen starvation. After filtering the data using Perseus, 1371 phosphorylation sites were identified and quantified in triplicate biological replicates that consisted of three independent N+ samples and N− samples for each time point.

### 3.1. Cluster Analysis

Hierarchical clustering was performed on the log-2 fold change (FC) of the label-free quantified intensity of phosphor-sites over time (i.e., log-2 LFQ[N−/N+], referred to as *LogetP*) [35,36] (Appendix A). Eight subclusters were created based on a similarity cutoff of cluster characteristics and were depicted as a heatmap. Moreover, a correlation matrix figure (Figure 1) was presented to show the correlation between subclusters in detail and to provide additional evidence for the heatmap results. When combining the information shown in the heatmap figure (Appendix A) and the correlation matrix figure (Figure 1), the result showed that the time points of 03 h, 06 h, and 12 h had similar pattern, then 48 h, and 10 d had similar pattern as well. 24 h was the critical point between two patterns.

### 3.2. Heatmap Subcluster Features

The top three functions for all the phosphor-sites affected by nitrogen starvation were gene expression (167 sites), protein synthesis (134 sites), and transport (112 sites). The temporal abundance profiles of these sites were broken down into eight heatmap subclusters, which were depicted in Figure 2 together with their most frequent protein functions. There were some noteworthy trends in clusters. For example, in cluster 1, the modification sites were primarily associated with gene expression, protein synthesis, and transport functions. Generally, there was an increase in modification sites from 3 h to 6 h, except for transport-related sites, which showed a significant decrease during this period. After 6 h, the trends for gene expression, protein synthesis, and transport-related modification sites became increasingly similar. In contrast, Cluster 5 displayed a consistent tendency from 3 to 48 h in the trends of modification sites related to gene expression, transport, and protein synthesis. However, a notable shift occurred at 48 h, characterized by a continuous increase in the transport category from 6 h to 48 h. Additionally, a significant shift was observed in the modification sites associated with gene expression and protein synthesis processes, with a decrease from 48 h to 10 days. This temporal dynamic alteration affected the overall trend in cluster 5. Furthermore, cluster 3 involved modification sites primarily linked to gene expression, protein synthesis, and other metabolic processes. The modification sites in this cluster showed a sustained decrease from 3 h to 12 h, followed by a rapid rise peaking at 48 h, and a significant decline from 48 h to 10 days. The modification sites related to these three functions followed similar patterns. Compared to cluster 3, cluster 4 showed transport as a novel functional category associated with the modification sites. The predominant functional categories were transport, gene expression, and protein synthesis processes. The associated sites showed a decline from 6 h to 12 h, followed by a significant increase from 24 h to 48 h.

### 3.3. Principal Component Analysis (PCA)

In the PCA (Figure 3), multiple principal components were identified. The first principal component (Dim 1) and the second principal component (Dim 2) were identified as the most significant patterns and directions of variability in the dataset, explaining 85.8% of the total variance. Dim 1 and Dim 2 accounted for 67.4% and 18.4% of the total variance, respectively. Furthermore, it shows that 24 h is a clear phase transition in sample behavior. Along Dim1 (Dim (67.4% variance), prior 24 h samples occupy opposite sides: the 03 h and 10 d points lie on the negative Dim 1 side, whereas the 06 to 12 h window already shifts to positive Dim 1, indicating an early activation phase. At the 24 h boundary, the sample remains on the positive Dim 1 side (around +1.2), confirming that the system has crossed into a distinct post 24 h state relative to baseline. Dim 2 (18.4% variance) further differentiates these phases: 6 to 12 h display positive Dim 2 values (a transient, early response), while the ≥24 h group diverges, with 24 h near the Dim2 midline and 48 h strongly negative (around −1.4), suggesting a reprogramming of pathways after 24 h. High cos 2 values (around 0.8–0.9) indicate that these separations are well represented in the two-dimensional space.

### 3.4. Analysis of Metabolic Pathways

To explore the regulatory functions of phosphor-sites under nitrogen starvation stress in *Nannochloropsis*, phosphorylation data were compared with proteomic data related to metabolic pathways such as photosynthesis, carbon metabolism, and nitrogen metabolism. The Venn diagrams showed the numbers of identified proteins and phosphorylated proteins of all samples corresponding to each metabolic pathway (Figure 4). In amino acid metabolism, 68 proteins and 46 phosphorylated proteins were detected, with 5 overlapping proteins: WD40 repeat-containing protein (WD40), prolyl endopeptidase (PE), amidophosphoribosyl-transferase (ATase), δ-1-pyrroline-5-carboxylate synthetase (P5CS), and amine oxidase (AOS). In carbon metabolism, 152 proteins and 28 phosphorylated proteins were identified, with 8 overlapping proteins: beta-galactosidase (β-GAL), phosphoenolpyruvate carboxykinase (PEPCK), trehalose phosphate synthase (TPS), exo-beta-glucanase (EXG), beta-glucosidase (β-GH), glycerol kinase (GYK), trehalose-6-phosphate synthase (T6P), and trehalose-phosphate synthase (AtTPS). In nitrogen metabolism, 21 proteins and 11 phosphorylated proteins were identified, with only 2 overlapping proteins: urease accessory protein (Ure) and glutamate ammonia ligase (GLUL). In lipid metabolism, 70 proteins and 41 phosphorylated proteins were detected, with 10 overlapping proteins: polyketide synthase (PKS), lecithin cholesterol acyltransferase (LCAT), beta-ketoacyl synthase (β-KS), lipid droplet surface protein (LDSP), acyl dehydratase (FabZ), hemolysin-related protein (HERP), propionyl-CoA carboxylase alpha chain (PCCA), glycerol-3-phosphate O-acyltransferase (GPAT), O-acyltransferase (OAT), and diacylglycerol acyltransferase family protein (DGAT). Among phosphorylated proteins, most were involved in lipid metabolism, whereas the fewest were found in nitrogen metabolism. During nitrogen depletion from 3 h to 10 days, proteins with identified phosphor-sites showed varying patterns: most proteins in amino acid metabolism displayed significant increases until 10 days, exemplified by ATase, although their phosphorylation levels remained stable. In carbon metabolism, the majority of protein abundances and phosphorylation levels were not significantly changed. In nitrogen metabolism, the abundance of most phosphorylated proteins was significantly increased. Both protein abundance and phosphorylation levels in lipid metabolism significantly increased until 10 days, as observed in LDSP, GPAT, and DGAT.

### 3.5. Analysis of the TOR Signaling Pathway

Many phosphor-sites and proteins were identified in association with the TOR signaling pathway. To better understand the relationship between quantitative proteome and phosphoproteomics analyses of the TOR signaling pathway, a figure was created (Figure 5) revealing 11 protein kinases and 6 phosphor-sites. Regarding phosphoinositide 3-kinase (PI3K), the protein abundance showed a significant increase (fold change > 1) at 48 h. However, the phosphorylation level was significantly downregulated at 48 h. For 3-phosphoinositide-dependent protein kinase 1 (PDK1), the protein abundance at 3 h significantly decreased (fold change < −1), but increased at 10 days. For adenosine 5’-monophosphate-activated protein kinase (AMPK), the protein abundance slightly increased at 12 h and then steadily decreased from 24 h to 10 days (fold change < 0.5). As a link between the tuberous sclerosis complex (TSC) and mechanistic target of rapamycin complex 1 (mTORC1), a total of 2 protein kinases and 3 phosphor-sites in the Ras homologue enriched in the brain (Rheb) were detected, including 2 different Ras-like proteins: Rab1A and Rab6A. The abundance of Rab1A decreased from 3 h to 6 h and slightly increased at 12 h. Three phosphorylation sites (S1, S2, S3) were identified for Rab6A. The abundance of Rab6A protein increased from 3 h to 10 days. Similarly, Rab6A (S1 and S2) phosphorylation levels increased over time from 3 h to 10 days. Site S1 showed the highest up-regulation at 10 days (fold change > 1), whereas site S3 showed the highest down-regulation at the same time (fold change < −1). In the Ras-related GTP-binding protein (Rag) GTPase complex, only one kinase was identified: Ras-related GTP-nuclear protein (RAN). Its protein level significantly increased at 3 h (fold change > 1) before decreasing at 6 h (fold change < −0.5). At the core of the TOR signaling pathway, the mTORC1 protein complex primarily comprises mTOR, RAPTOR, and DEPTOR. One phosphor-site was identified in RAPTOR, and one serine/threonine-protein kinase (AKT) was found in mTOR. The phosphorylation of RAPTOR was significantly downregulated at 10 days (fold change < −1). However, AKT in mTOR was significantly upregulated at 10 days (fold change > 1). Three distinct protein kinases were detected that putatively inhibit the mTORC1 protein complex: FKBP-type peptidyl-prolyl cis-trans isomerase (NbFKPPIase), FKBP Prolyl Isomerase 2 (FKBP2), and 12-kDa FK506-binding protein (FKBP12). Notably, FKBP2 showed significant changes over time. Its abundance significantly decreased after 3 h (fold change < −1). FKBP12 showed a consistent increase (fold change < 0.5) from 24 h to 10 days. Additionally, NbFKPPIase exhibited a slight increase in protein levels at 12 h (fold change < 0.5). In the 70-kDa Ribosomal protein S6 kinase (p70S6K), serine-threonine kinase receptor-associated protein (STRAP) was identified. At the protein level, the abundance of STRAP showed a consistent increase (fold change < 0.5) from 24 h to 10 days. Regarding ribosomal protein S6 (RPS6), its abundance consistently declined from 3 to 48 h before slightly increasing after 10 days.

In summary, the comparison of protein abundance with phosphorylation levels in the TOR signaling pathway revealed that phosphorylation played a prominent role in regulating protein function. Unlike most proteins in the TOR signaling pathway, which remained relatively stable until 10 days, their phosphorylation sites underwent dynamic changes immediately after nitrogen depletion, as exemplified by significant changes in the phosphorylation status of Rheb observed after 3 h.

## 4. Discussion

### 4.1. Adaptation to Nitrogen Starvation

Clustering analysis of Phosphoproteomics(PMO) data revealed the distinct physiological phases of *Nannochloropsis* under nitrogen deficiency stress, suggesting a functionally compartmentalized adaptation strategy. Specimens collected via continuous sampling were categorized into two primary subclasses, early-to-mid (3–24 h) and late (48 h–10 d) phases, demarcated at 24 h. This PMO-based phasing intriguingly diverges from our previous three-stage model derived from transcriptomics and proteomics [32], highlighting that phosphorylation dynamics offer a unique temporal perspective on the stress response (Figure 1). While specific phosphorylation patterns varied temporally (Figure 1), the key insight is the immediate and unique response at 3 h, positioning phosphorylation as a primary sensor mechanism. The prominence of the 3 h time point in our PMO data, in contrast to our former research omics data like proteomics and metabolomics data, strongly implicates phosphorylation as a critical component of the initial cellular alarm system, triggering early adaptive maneuvers before substantial changes in transcript or protein abundance occur. The antagonistic phosphorylation patterns observed in clusters 1 and 5 between 3 h and 48 h underscore a sophisticated regulatory logic. This is not a simple on/off switch but a concerted re-wiring of the cellular state, where simultaneous up- and down-regulation of phosphorylation across different protein cohorts fine-tunes metabolic pathways. This nuanced control mirrors mechanisms observed in other eukaryotes where kinase/phosphatase networks orchestrate rapid stress adaptations. Our previous findings on cellular behavior alterations at 48 h are consistent with these findings, thereby providing additional evidence for the crucial role of phosphorylation in regulating cellular metabolic adaptation. However its true novelty lies in revealing the initiation of these adaptations hours before they manifest at the proteome level. The case of long-chain acyl-CoA synthetase (ACSL) serves as a paradigm for phosphorylation-mediated metabolic control. The phosphorylation abundance of this protein significantly increased at 3 h and then decreased at 48 h. Preliminary studies have indicated that within the first 48 h of nitrogen deprivation in *Nannochloropsis*, TAGs predominantly originate from the Kennedy pathway, which depends on ACSL, highlighting this pathway as the major source of TAGs during the early stages of nitrogen starvation. This indicated that the phosphorylation state of ACSL becomes a critical regulatory node. The temporal inverse correlation between ACSL phosphorylation and TAG accumulation, despite stable protein levels, powerfully argues that phosphorylation directly regulates the ACSL activity and thus TAG flux through the Kennedy pathway. This proposed inhibitory phosphorylation is a well-established regulatory mechanism for metabolic enzymes in other systems, such as the phosphorylation and inactivation of acetyl-CoA carboxylase (ACCase) in mammals, which similarly controls carbon commitment to lipid synthesis. After 48 h, while the phosphorylation status showed minimal change, a substantial increase in both ACSL abundance and TAG levels was observed. This reveals an elegant, two-tiered regulatory strategy: an immediate, post-translational brake applied via phosphorylation is later superseded by a sustained, gene expression accelerator that increases the enzyme abundance. This bi-layered model explains the kinetic delay in maximal TAG production. Similarly, sn-1-diacylglycerol lipases (Sn-1-DAGL), cobalamin synthesis protein (CobW protein), and prolyl endo-peptidase (PE) further reinforce the concept of an extensive phospho-regulatory network governing the nitrogen stress response. This finding highlights the crucial regulatory function of phosphorylation and dephosphorylation events targeting the same protein, which plays a key role in modulating the activity of *Nannochloropsis* during the distinct stages of its adaptation to nitrogen limitation. Consequently, the differential activation states of these proteins orchestrate the intricate metabolic processes underlying the physiological adjustments of *Nannochloropsis* to nutrient stress, further emphasizing the dynamic interplay between phosphorylation-mediated signaling and metabolic regulation.

Our previous studies revealed that components of photosynthetic system are repurposed to provide carbon skeletons for fatty acid biosynthesis in *Nannochloropsis* under nitrogen stress [37,38,39]. Nitrogen stress not only reduces the abundance of chlorophyll and carotenoid proteins but also decrease chloroplast lipid content [25,26,40,41]. Our previous proteomics revealed a specific decreases in PSI-associated light-harvesting complex (LHCs), contrasting with mRNA data, and the general reduction in activity of both PSI and PSII LHCs [40], hinting at post-transcriptional and post-translational control mechanisms. Our PMO data now offer a novel mechanistic perspective for the observed decline in photosynthetic rate, implicating the phosphorylation status of photosystem II D2 protein (PSII D2). We observed a significant increase in phosphorylation at its specific sites at 24 h and 10 d under nitrogen starvation. Fv/Fm, which indicates the degree of damage to the photo-systems, gradually decreased over the entire nitrogen deficiency period according to previous research [42]. Critically, studies in higher plants establish a precedent where phosphorylation of PSII core proteins, including D2, slows their degradation under photo-inhibitory conditions [26,42], a conserved mechanism that protects the PSII complex from excessive damage. Our PMO data combined with the former protein data strongly suggests that this phospho-protective mechanism is functionally conserved in *Nannochloropsis* under nitrogen stress. Furthermore, this reversible phosphorylation mechanism provides a plausible explanation for the rapid recovery of photosynthesis upon nitrogen resupply, since phosphorylation is a reversible process that can occur rapidly [43].

### 4.2. TOR Signaling Under N-Limiting Conditions

The Target of Rapamycin (TOR) signaling pathway is an essential component of the intricate mechanisms underlying energy and nutrient homeostasis in eukaryotic organisms. This fundamental pathway is conserved across a broad phylogenetic spectrum of yeast, mammals, and plant species. The integration of our proteomic [32] and current PMO datasets provides compelling evidence that phosphorylation is a pivotal mechanism modulating the response to nitrogen starvation in *Nannochloropsis* through modulation of the TOR signaling cascade. Our phosphoproteomic analysis revealed dynamic and significant changes in the phosphorylation of core TOR signaling components (Figure 3), implying a rapid rewiring of this central regulatory network upon nitrogen withdrawal. The fact that these phosphorylation changes occurred much earlier than any significant alterations in the abundance of the TOR pathway proteins themselves underscores phosphorylation as the primary, fast-acting regulatory layer for TOR signaling during the early and mid-phase nitrogen stress response. Given nitrogen’s status as a key growth signal, the conservation of nitrogen-dependent TOR regulation in *Nannochloropsis* is unsurprising yet significant, as it details this mechanism in a bio-technologically relevant microalga [31].

### 4.3. TOR-Mediated Regulation of Fatty Acid Synthesis

Building upon prior research [44], it has been demonstrated that the phosphorylation status of TOR pathway-associated proteins, i.e., a decrease at 3 h followed by an increase, correlates with the corresponding low abundance of TAG levels at 3 h and an increase after 6 h in *Nannochloropsis*. This alignment establishes a causal link between TOR signaling dynamics and the regulation of fatty acid biosynthesis in microalgae [45]. Former research found that transcriptome and proteome levels of lipid synthesis proteins change during nitrogen depletion. Our PMO data strongly indicated the TOR as the upstream orchestrator of these events. Furthermore, the correlation between TOR pathway phosphorylation and TCA cycle activity suggests an additional regulatory axis where TOR might govern carbon partitioning, directing flux away from central carbon metabolism towards lipid storage. This multifaceted regulation echoes the mechanism in mammalian systems, where TORC1 controls fatty acid synthesis transcriptionally via Sterol Regulatory Element-Binding family proteins (SREBP), ACSL and Stearoyl-CoA Desaturase 1 (SCD1) to regulate the de novo synthesis of fatty acids [46]. The conservation of TOR-mediated lipid regulation likely stems from TOR’s primordial role as a nutrient sensor and the fundamental importance of lipid metabolism for energy storage and membrane integrity across eukaryotes. Consequently, a key objective was to dissect the temporally distinct roles of TOR during early (0-48 h) versus prolonged (10 d) nitrogen deficiency.

Previously, it has been shown that suppressing fatty acid synthesis under nutrient-limited conditions aligns with the cellular economy necessitated by such stresses. TOR, as a master metabolic regulator, is central to implementing this economy by repressing anabolic pathways like de novo fatty acid synthesis in response to nutrient scarcity, thereby maintaining homeostasis [31,45]. However, the observed promotion of lipid accumulation under prolonged nitrogen deprivation presents a paradox. This compels us to refine the model: How does TOR’s regulatory role evolve between short-term and long-term nitrogen deficiency to enable this shift?

Studies in other industrial models of the microalgae *Chlorella sorokiniana* and *Chlamydomonas reinhardtii* revealed that nitrogen starvation decreased the activity of the TOR complex [44]. This cross-species consistency underscores the general importance of TOR in microalgal nitrogen responses but leaves the temporal dynamics and specific mechanisms in *Nannochloropsis* unresolved. Our PMO analysis suggests a two-pronged regulatory model for TOR in *Nannochloropsis*: it influences lipid synthesis not only through potential transcriptional/post-translational control of ACCase/FAS but also by potentially modulating carbon partitioning at the TCA cycle gateway, thereby orchestrating the lipid synthesis program in response to nitrogen status. In summary, according to previous studies and our results, TOR employs a temporally layered, multifaceted regulatory strategy to address environmental challenges such as nitrogen starvation, ensuring metabolic adaptation of lipids within the cell. Several prior studies have similar observations to ours from multiple omics studies on transcriptomics, proteomics, and lipidomics. Unfortunately, a key limitation is that they did not analyze the changes in the TOR signaling pathway in response to long-term (more than 48 h) nitrogen deprivation in algae, leaving the mechanism for sustained TAG accumulation under long-term starvation unresolved. By extending to 10 days, our study addresses this gap. Our prior research works showed that most genes involved in de novo fatty acid synthesis are downregulated in the short term. However, pathways associated with TAG synthesis, particularly the eukaryotic Kennedy pathway and the phospholipid diacylglycerol acyltransferase (PDAT)-mediated alternative route are unregulated at the transcriptional level [47]. Integrating our new long-term PMO data with prior multi-omics data yields a coherent model: sustained nitrogen stress triggers a strategic metabolic re-routing. Carbon precursors are diverted from protein/carbohydrate metabolism towards glycerolipid synthesis, primarily by stimulating TAG assembly pathways, even while de novo fatty acid synthesis remains suppressed. Consequently, the net increase in TAG accumulation under prolonged stress is driven not by accelerated de novo FA creation, but by reinforced assembly of existing or alternatively sourced fatty acids into TAG. This result aligns with emerging notions in microalgal lipid biology that TAG accumulation of *Nannochloropsis* under nitrogen stress is governed predominantly by the assembly step rather than the fatty acid synthesis step [48].

### 4.4. Study Limitations

It should be noted that this study also has some limitations. First, although we utilized three biological replicates, which is common in microbial omics studies, more bioreplication would undoubtedly increase statistical accuracy and strengthen the generalizability of our conclusions. Second, although we endeavored to minimize the impact of technical variations through rigorous data pre-processing (including normalization and the collection of phospho- and proteome sample from same time points), we cannot entirely rule out that residual batch effects might subtly influence the integrated interpretations. For instance, the phosphoproteomic and proteomic datasets were acquired in separate MS batches rather than concurrently, a factor that could have introduced batch effects into the integrated comparisons. Despite these limitations, the findings from this integrated analysis provide a valuable foundation for understanding the phospho-regulatory network in microalgae under nitrogen stress.

## 5. Conclusions

In conclusion, our study establishes phosphorylation-mediated signaling as a central regulatory paradigm through which *Nannochloropsis oceanica* dynamically reprograms its metabolic network in response to nitrogen deprivation. We reveal a biphasic regulatory architecture—comprising early adaptive adjustments and late survival strategies—that fundamentally reorganizes cellular priorities over time. Notably, phosphorylation exerts direct control over lipid synthesis enzymes, representing a rapid post-translational mechanism that precedes and complements transcriptional responses. The TOR signaling pathway emerges as a master conductor of this metabolic rewiring, differentially gatekeeping carbon flux between primary metabolism and storage lipid assembly depending on stress duration. This temporal regulatory system resolves the paradox wherein nitrogen starvation initially suppresses yet ultimately stimulates the lipid synthesis of Nannochloropsis. We propose a novel model wherein phosphorylation networks enable metabolic flexibility by toggling between energy conservation and lipid accumulation modes. These mechanistic insights provide biotechnological leverage points of synthetic biology, such as mutating the phosphorylation modification site to enhance lipid yields. The elucidated TOR regulatory logic further suggests strategies for synchronizing lipid accumulation phases in industrial cultivation, like balancing biomass accumulation and lipid synthesis. By mapping the phosphorylation landscape underlying metabolic transitions, this work provides both a conceptual framework for understanding N- stress adaptation and profiles the potential targets for strain improvement.

## Figures and Tables

**Figure 1 biology-14-01599-f001:**
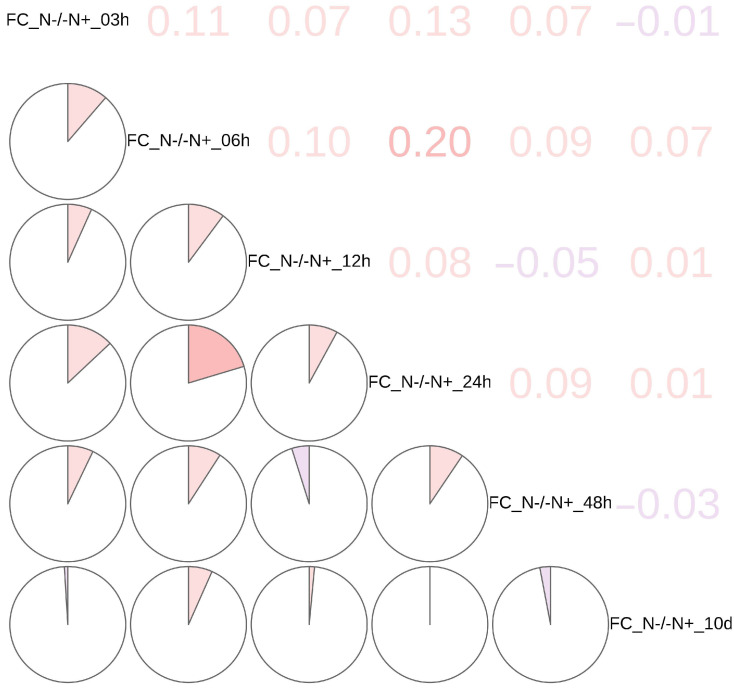
Phosphoproteomics analysis of N−/N+ conditions and correlation matrix. Correlation matrix shows Pearson correlation of log-2 fold change (FC) across the nitrogen starvation time course. Color intensity indicates correlation strength (red: positive, blue: negative).

**Figure 2 biology-14-01599-f002:**
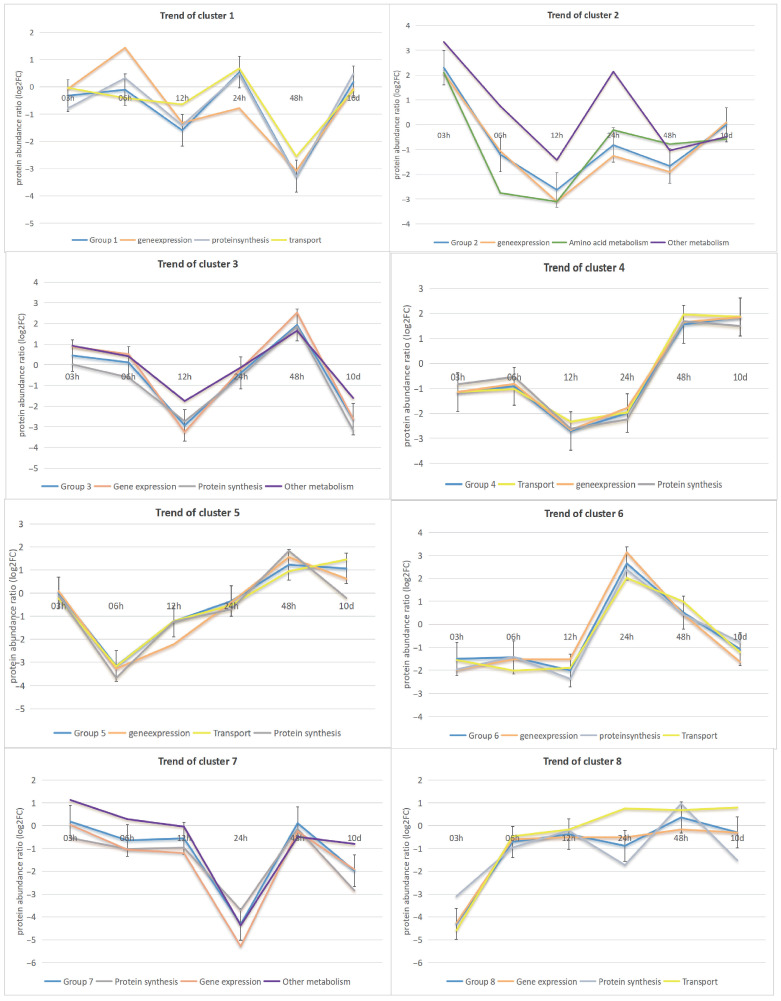
Comparison of temporal trend in protein function for 8 subclusters. This figure shows the most prevalent protein functions associated with altered phosphorylation sites, as inferred from gene annotation data through homology searches using BLASTP (version 2.0) against the NCBI Non-redundant protein sequences (nr) database. The subcluster delineates the temporal trends of these protein functions over time. In each subcluster, the blue line represents the average temporal trend of the subcluster.

**Figure 3 biology-14-01599-f003:**
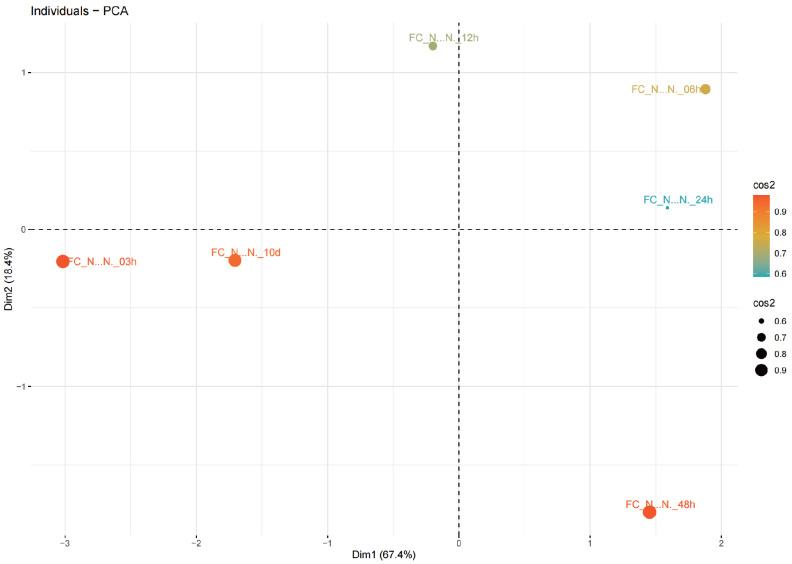
PCA analysis of the average log2 LFQ(N-/N+) of phosphoprotein samples. Color intensity and size of the dot (cos2) are used to present the contribution of average log2 LFQ(N−/N+) of each group of phosphoprotein samples within all phosphoprotein samples.

**Figure 4 biology-14-01599-f004:**
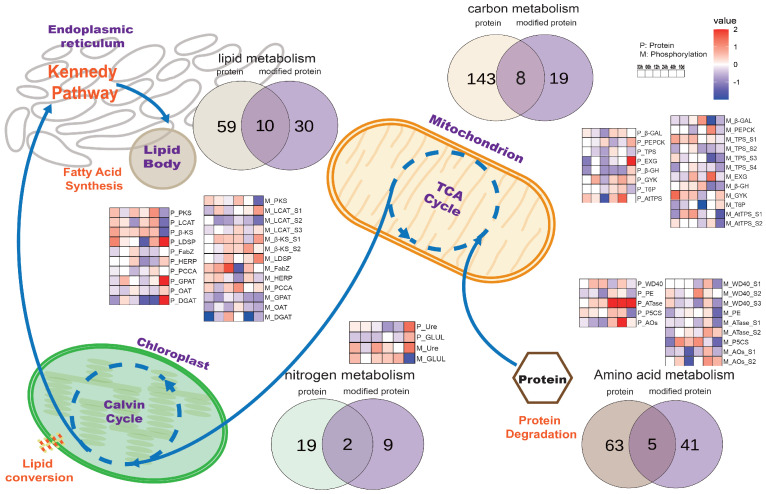
Overview of metabolic pathways. It illustrates the integration of phosphorylation and proteomics data associated with key metabolic pathways, including photosynthesis, carbon metabolism, nitrogen metabolism, and lipid metabolism. The displayed Venn diagrams summarize the number of unique and shared proteins and phosphorylated proteins across these pathways. Heatmaps present the temporal trends of the abundance of proteins and phosphorylation modification sites for overlapping proteins, from the initial 3 h to the final 10 days. Each protein has a prefix, where “P_” means unmodified and “M_” means phosphorylated protein, followed by an acronym for the protein name. Color intensity is used to denote the ratio log2LFQ(N−/N+)of protein to phosphorylation modifications, respectively.

**Figure 5 biology-14-01599-f005:**
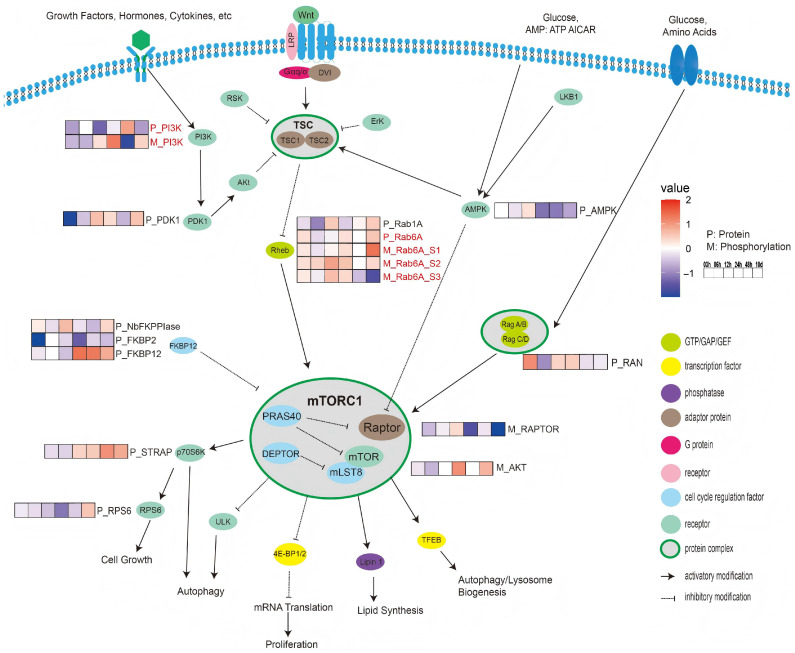
Overview of TOR signaling pathway. It displays the integration of phosphorylation and proteomic data within the context of the TOR signaling pathway. Heatmaps present temporal trends in the abundance of phosphorylation modification sites with node proteins of TOR signaling pathway. Each node protein is differentiated by color that corresponds to its specific category. Every protein has a prefix, where “P_” indicates unmodified protein and “M_” indicates phosphorylated protein, followed by an acronym for protein name. Color intensity is used to denote the ratio log2LFQ(N−/N+) for protein and phosphorylation modifications, respectively.

## Data Availability

The mass spectrometry proteomics data have been deposited to the ProteomeXchange Consortium (https://proteomecentral.proteomexchange.org) via the iProX partner repository with the dataset identifier PXD055733.

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
