# Peer review of "Integrative Multi-Omics Analyses Reveal the Global Regulation Network of the Microalga *Nannochloropsis oceanica* Under Nitrogen Stress Adaptation"

_biology, 2025, doi:10.3390/biology14111599_

Round 1
Reviewer 1 Report
Comments and Suggestions for Authors
Title:
Integrative multi-omics analyses reveal the global regulation network of the microalga Nannochloropsis oceanica under nitrogen stress adaptation
General comments:
This study provides a comprehensive phosphoproteomic analysis of Nannochloropsis oceanica under nitrogen stress, integrating these new data with previously published transcriptomic and proteomic datasets. However, several issues in each section of the manuscript need to be addressed, as outlined in the following sections, to enhance the manuscript.
Abstract:
The abstract effectively summarizes the study’s findings but tends to overstate the conclusiveness of the results. Statements such as “Results showed……” and “The TOR signaling pathway involved 11 proteins, indicating phosphorylation’s role…….” are descriptive but lack the nuance expected in a complex multi-omics study.
The concluding phrase that the study “offers new insights” is also too general. The main contributions should be specified and described, such as the proposed dual regulatory role of phosphorylation or the identification of early regulatory events preceding proteomic changes.
Introduction:
The rationale for emphasizing phosphorylation could be more adequately described. Although the manuscript mentions that inconsistencies between transcript and protein levels inspired this investigation, however, it does not clearly define specific hypotheses on how phosphorylation might explain these differences or directly influence metabolic regulation.
Materials and Methods:
The manuscript mentioned that the algal samples originated from the same batch used in a previous study, however, the definition of biological replicates in this phosphoproteomic experiment remains unclear. It should be specified whether the replicates represent technical repeats of the same harvested biomass or independent biological cultures, as this point is critical for interpreting the results.
The description of the statistical analysis is inadequate. Employing a t-test without applying any correction for multiple comparisons across thousands of phosphosites represents a major limitation. It increases the likelihood of false-positive results.
The manuscript does not clearly describe the approach used to integrate the new phosphoproteomic data with the previously published proteomic and transcriptomic datasets. It remains uncertain whether this integration involved a computational re-analysis or merely a side-by-side comparison.
Results:
The results section contains a substantial amount of data but tends to be overly descriptive, which causes difficulty for the reader.A more hypothesis-driven presentation that emphasizes the key findings rather than listing all observations would greatly improve clarity and impact.
The interpretation of the clustering analysis appears inconsistent. The manuscript notes that the 24 h and 10 d samples are similar, as are the 6 h and 12 h samples in Figure 1b, whereas the heatmap
Figure 1a and the disucussion at lines 295-305 indicate the more complex and distinct pattern for the 24 h sample. This discrepancy should be clarified to ensure consistency between the text and the figures.
Figure 2 presents an excessive amount of information, which reduces its clarity and interpretability. The trends within individual functional categories are difficult to distinguish from the overlapping lines. A more focused presentation highlighting only the most relevant clusters would make the figure more informative and easier to interpret.
Discussion:
The discussion section often replicates the results rather than offering a deeper critical interpretation or integrative analysis.
The abbreviation “PMO” appears in the discussion section without being defined at its first occurrence, and should be clarified for the reader.
The manuscript offers a limited comparison of its findings with previous studies in microalgae or model organisms. For e.g., the proposed phosphorylation-based regulation of ACSL and the PSII D2 protein is intriguing but would be more convincing if supported by a discussion of similar regulatory mechanisms reported in other systems.
The discussion does not adequately acknowledge key limitations of the study, including the small number of biological replicates and the possible influence of batch effects in the multi-omics integration.
Conclusion:
The conclusion mainly restates the results and discussion rather than emphasizing the key conceptual advances of the study. It should be revised to clearly articulate the main contributions.
Additionally, the final statement about 'valuable insights for future biotechnological applications' is too general and should specify the potential applications envisioned.
Reviewer 2 Report
Comments and Suggestions for Authors
The manuscript presents an ambitious and well-structured integrative multi-omics study of Nannochloropsis oceanica under nitrogen stress, but several issues should be addressed for clarity and rigor. In the Abstract, claims of direct regulation by phosphorylation should be tempered, as the study provides correlative rather than functional evidence. The Introduction would benefit from a sharper statement of novelty, since nitrogen stress–induced lipid accumulation and TOR involvement are already known; it should highlight what phosphoproteomics specifically adds. In the Methods, more detail is needed on experimental replicates, normalization of phosphoproteomics data, and statistical thresholds for site significance to ensure reproducibility. The Results and Discussion contain valuable insights, but causal language (phosphorylation inhibits ACSL activity) should be rephrased to reflect correlation, unless direct enzymatic assays are provided. The clustering analysis is described clearly, but mechanistic interpretation sometimes overextends beyond the data (TOR-mediated carbon flux regulation without metabolic flux validation). Photosynthesis regulation is an interesting angle, but connections between phosphorylation and functional outcomes (Fv/Fm decline) should be clarified. The Conclusions should be more concise, avoiding overgeneralized statements about biotechnology and instead specifying potential applications (strain engineering targets, stress-resilient lipid productivity). Across the manuscript, integrating figures more tightly with the text, acknowledging limitations (short sampling window, lack of mutant or inhibitor validation), and specifying what is novel compared to previous transcriptomic/proteomic studies would substantially improve clarity and impact.

Reviewer 3 Report
Comments and Suggestions for Authors
The manuscript entitled “Integrative multi-omics analyses reveal the global regulation network of the microalga Nannochloropsis oceanica under nitrogen stress adaptation” is devoted to Nannochloropsis oceanica lipids accumulation as under nitrogen-depleted and replete conditions. To my mind this manuscript is corresponding to the aims and scopes of the Biology journal.. I am ready to recommend it for publication, due to the few comments below.
This is an important study carried out at a modern scientific level. However, it should be noted that the authors rushed through the manuscript and failed to pay due attention to various formal and other aspects.
The abstract is too brief, contains unexplained abbreviations, and does not always correspond to the scientific style of writing. It contains rather odd expressions, such as "phosphoproteomic data." The abstract needs to provide introductory information for readers unfamiliar with the topic of the study.
While the introduction is well written and provides a detailed background, it lacks a clearly stated objective for this study.
Section 2.1 should be expanded upon, indicating the source of the strain.
Sections 2.2 and 2.3 are hastily written; they need to be described in slightly more detail, paying attention to the indices in the formulas (as throughout the rest of the text).
Do the authors really think the reader will see anything in Figure 1.a? It should be moved to supplementary materials where it can be downloaded and enlarged.
In Figure 1b, the colored numbers are illegible. The description of Figure 1 is rather mechanical and sparse.
Figure 2b should be supplemented with error bars, if possible. The formatting of the figures in Excel is not very good.
Figures 4 and 5 are excellent, but not all parts (heat maps) are easy to read. Perhaps they should be separated into separate illustrations?
The authors also wrote a very well-written conclusion. It clearly demonstrates the vast amount of material they processed and the high value of the work.
Round 2
Reviewer 1 Report
Comments and Suggestions for Authors
I am satisfied with the revised version and have no further queries.
Reviewer 2 Report
Comments and Suggestions for Authors
The authors have made substantial and thoughtful revisions in response to the previous review comments. The revised manuscript is now clearer, more focused, and better organized, with improved figures and a strengthened discussion of the TOR signaling pathway and its regulatory role in nitrogen-stress adaptation and lipid metabolism. The integration of phosphoproteomic, transcriptomic, and metabolomic analyses is presented more coherently, and the methodological details are now easier to follow. Overall, the authors have successfully addressed prior concerns, enhanced the scientific rigor and readability of the work and provided valuable and well supported contribution to understanding phosphorylation mediated regulation in Nannochloropsis oceanica.
Reviewer 3 Report
Comments and Suggestions for Authors
To my opinion, the authors have significantly revised the manuscript and taken into account all my comments. I am ready to recommend the manuscript for publication in this form.